# Deep Learning Application for Crop Classification via Multi-Temporal Remote Sensing Images

**Qianjing Li** [1,2], **Jia Tian** [1,3,*] **and Qingjiu Tian** [1,2]

1 International Institute for Earth System Science, Nanjing University, Nanjing 210023, China
2 Jiangsu Provincial Key Laboratory of Geographic Information Science and Technology, Nanjing University, Nanjing 210023, China
3 School of Instrumentation and Optoelectronic Engineering, Beihang University, Beijing 100191, China
* Correspondence: tianjia@nju.edu.cn

**Abstract:** The combination of multi-temporal images and deep learning is an efficient way to obtain accurate crop distributions and so has drawn increasing attention. However, few studies have compared deep learning models with different architectures, so it remains unclear how a deep learning model should be selected for multi-temporal crop classification, and the best possible accuracy is. To address this issue, the present work compares and analyzes a crop classification application based on deep learning models and different time-series data to exploit the possibility of improving crop classification accuracy. Using Multi-temporal Sentinel-2 images as source data, time-series classification datasets are constructed based on vegetation indexes (VIs) and spectral stacking, respectively, following which we compare and evaluate the crop classification application based on time-series datasets and five deep learning architectures: (1) one-dimensional convolutional neural networks (1D-CNNs), (2) long short-term memory (LSTM), (3) two-dimensional-CNNs (2D-CNNs), (4) three-dimensional-CNNs (3D-CNNs), and (5) two-dimensional convolutional LSTM (ConvLSTM2D). The results show that the accuracy of both 1D-CNN (92.5%) and LSTM (93.25%) is higher than that of random forest (~91%) when using a single temporal feature as input. The 2D-CNN model integrates temporal and spatial information and is slightly more accurate (94.76%), but fails to fully utilize its multi-spectral features. The accuracy of 1D-CNN and LSTM models integrated with temporal and multi-spectral features is 96.94% and 96.84%, respectively. However, neither model can extract spatial information. The accuracy of 3D-CNN and ConvLSTM2D models is 97.43% and 97.25%, respectively. The experimental results show limited accuracy for crop classification based on single temporal features, whereas the combination of temporal features with multi-spectral or spatial information significantly improves classification accuracy. The 3D-CNN and ConvLSTM2D models are thus the best deep learning architectures for multi-temporal crop classification. However, the ConvLSTM architecture combining recurrent neural networks and CNNs should be further developed for multi-temporal image crop classification.

**Keywords:** crop type classification; deep learning; multi-temporal; remote sensing

## 1. Introduction

Detailed and accurate information on crop-type cultivation is essential for developing economically and ecologically sustainable agricultural strategies in a changing climate, and for satisfying human food demands [1]. Multi-temporal remote sensing (RS) images acquired throughout the growing season provide an effective method for acquiring crop cover information over large areas [1,2]. Multi-temporal images can be used to distinguish crop growth states and the phenological characteristics of crops. In addition, they provide enriched features that allow more complex and stable crop classification tasks. They have thus seen wide use in the field of agricultural RS [3,4].

Two main strategies are available for multi-temporal crop classification. The first strategy is to stack multi-temporal images by time sequence and classify them with classifiers

such as support vector machine (SVM), random forest and maximum likelihood [5,6]. However, this approach does not model temporal correlations and uses features independently, ignoring possible temporal dependencies [6,7]. Most classifiers such as SVM rely heavily on features that are not designed for time-series data, making it difficult to exploit any inherent time-series variability features. In addition, the stacked images increase redundancy and lead to the dimensionality catastrophe with increasing time-series length, which negatively affects classification performance [6,8]. The second strategy is to obtain new images from reflectance images by using spectral indices, such as the normalized difference vegetation index (NDVI) and the enhanced vegetation index (EVI), and then construct time-series data to reveal the temporal pattern of the different features. With this method, crops and other vegetation are classified with high accuracy. However, the classification results of this method are limited strictly by the number of images in the time-series. If the number is too small, then the temporal pattern has little effect on classification performance [8]. In addition, manual feature engineering based on human experience and prior knowledge is essential with this approach, which increases the complexity of processing and computation [7,9]. Moreover, the construction of VIs based on the specific spectral features ignores other spectral bands, which in turn affects the classification performance.

Current multi-temporal RS images are multi-spectral, multi-temporal and multi-spatial. In multi-temporal images, crops are represented via variations in temporal, spectral, and spatial features. These features can be comprehensively included in four-dimensional (4D: time, height, width, and band) data that require classification models to learn and represent temporal, spectral, and spatial features. Multi-temporal images thus pose new challenges to the models used for data processing, so integrating multi-temporal images and continuously improving crop classification accuracy requires continued attention.

Deep learning is a breakthrough technique in machine learning that outperforms traditional algorithms in terms of feature extraction and representation [5–7], which has led to its application in numerous RS classification tasks [8–10]. Convolutional neural networks (CNNs) produce more accurate results than other models in most RS image classification problems [8,9,11]. The one-dimensional CNN (1D-CNN) model is commonly used to extract spectral features from hyperspectral images or temporal features from time-series images, providing an effective and efficient method for crop identification in time-series RS images [12]. The CNN learning process is computationally efficient and insensitive to data shifts such as image translation, allowing CNN models to recognize image patterns in two dimensions (2D) [13]. Three-dimensional (3D) CNN models use the spatial, temporal, and spectral information in multi-temporal images, and therefore are widely used in multi-temporal crop classification [11,14]. Long short-term memory (LSTM), a variant of recurrent neural networks (RNNs), is a natural candidate to represent temporal dependency over various temporal periods with gated recurrent connections [9,15]. LSTM models have been widely used for multi-temporal crop classification because they can analyze sequential data [9,16,17]. For multi-temporal crop classification, both CNN and RNN provide more accurate results than machine learning and traditional classification [5,9,11]. However, various deep learning architectures produce different results when applied to multi-temporal crop classification, feature learning and representation of crop spectral, spatial, and temporal information.

Convolutional LSTM (ConvLSTM) is a type of RNN with internal matrix multiplication replaced by convolution operations [18]. ConvLSTM, integrating both LSTM and CNN structures, shows unexpected adaptability to multi-temporal images [19–21]. However, due to the prevalence of CNNs and RNNs and the requirement for higher data dimensions, the ConvLSTM model is less commonly used in multi-temporal crop classification. Nevertheless, the potential of the ConvLSTM model deserves further exploration.

To summarize, multi-temporal images pose a new challenge to classification models in terms of data processing and feature extraction, but also open new opportunities for using data-driven deep learning to classify RS images. In this work, we use multi-temporal Sentinel-2 RS images as input data, and analyze the advantages of using such data and the

structural advantages of various deep learning models. This research investigates (1) the possibility of using multi-temporal images for more accurately classifying crops; (2) the contribution of spectral, temporal, and spatial information to multi-temporal crop classification; and (3) the potential and requirements of using deep learning for multi-temporal crop classification. We also (4) search for a feasible and suitable deep learning model that provides optimum classification accuracy from multi-temporal images. Although such deep learning models have long been used for RS applications, this work compares and analyzes multi-temporal crop classification based on the deep learning architectures of CNN, LSTM, and ConvLSTM.

## 2. Materials

### 2.1. Study Area

The study area, Norman county, is located in northwestern Minnesota (Figure 1), which is a highly productive agricultural state in the United States. Minnesota is in the Great Plains of the central United States, and agricultural land covers the vast majority of the study area. The continental climate of the region is cold in the winter and hot and humid in the summer, with 600 mm/year of precipitation. The highest temperatures occur in July, and the lowest in January, with an average of 197 sunny days per year. The climatic and temperature conditions make single-season crop cultivation the main cropping system. The major crops in this region are corn, soybeans, sugarbeets, and spring wheat, which are planted in about 89% of the study area. Corn begins being planted at the end of April, matures in September, and is harvested through October. Soybeans are planted in May and harvested from mid-September through the end of October. Spring wheat is sown in early April and harvested from mid-July through August. Sugarbeets are planted in mid-April, mature in September, and are harvested by the end of October.

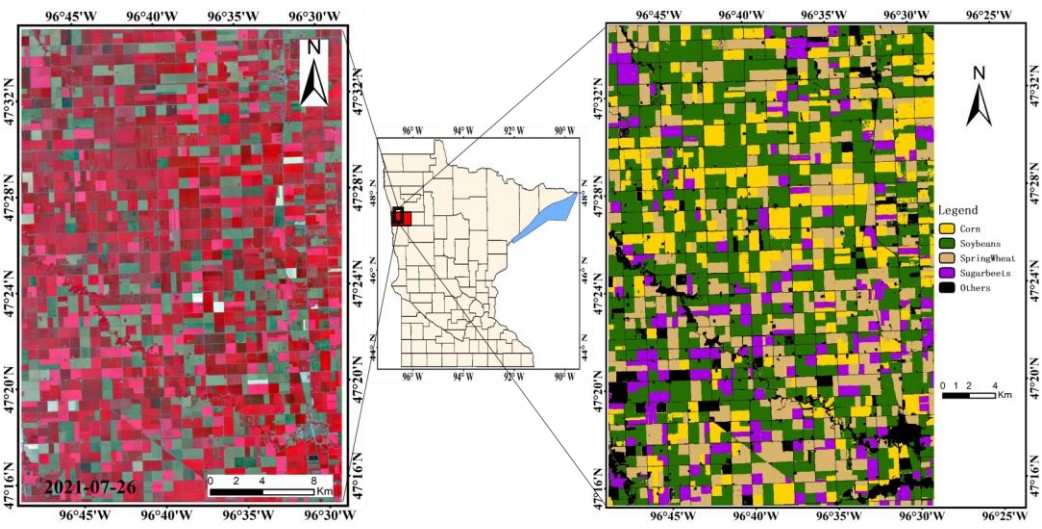

**Figure 1.** False color image and the Cropland Data Layer (CDL) of study areas.

### 2.2. Data

#### 2.2.1. Remote Sensing Images

Sentinel-2 images were downloaded from the Sentinel Hub (https://www.sentinel-hub.com/ (accessed on 28 October 2022)). Cloud-free images from April 2021 to October 2021 were selected to encompass the entire crop growing season. A total of 13 Sentinel-2 images (Tables 1 and 2) were selected as the main input data of the experiment. Data preparation involved stacking and resampling the 20 m spectral bands to 10 m and the removal of the coastal band, water vapor, and the cirrus band, accomplished through the Sentinel Application Platform (SNAP).

**Table 1.** Spectral bands of Sentinel-2 images.

| Band Names | Spectral Band | Central Wavelength (nm) | Band Names | Spectral Band | Central Wavelength (nm) |
|---|---|---|---|---|---|
| Blue | B2 | 490 | Red-Edge | B7 | 775 |
| Green | B3 | 560 | NIR | B8 | 842 |
| Red | B4 | 665 | NIR | B8a | 865 |
| Red-Edge | B5 | 705 | SWIR | B11 | 1610 |
| Red-Edge | B6 | 740 | SWIR | B12 | 2190 |

**Table 2.** Acquisition time of Sentinel-2 images.

| Day of Year (DOY) | Acquisition Time | Day of Year (DOY) | Acquisition Time |
|---|---|---|---|
| 112 | 22 April 2021 | 230 | 18 August 2021 |
| 137 | 17 May 2021 | 235 | 23 August 2021 |
| 150 | 30 May 2021 | 242 | 30 August 2021 |
| 165 | 14 June 2021 | 257 | 14 September 2021 |
| 192 | 11 July 2021 | 270 | 27 September 2021 |
| 207 | 26 July 2021 | 295 | 22 October 2021 |
| 225 | 13 August 2021 | | |

2.2.2. Training and Validation Samples

The Cropland Data Layer (CDL) is a crop-type distribution product published by the United States Department of Agriculture and the National Agricultural Statistics Service. The 2021 CDL (Figure 1) for Norman County has a spatial resolution of 30 m, and was obtained from the CropScape website portal (https://nassgeodata.gmu.edu/CropScape/ (accessed on 20 October 2022)). Although the CDL is not the absolute ground truth, it is the most accurate crop-type product available, especially for corn and soybeans, with over 95% accuracy [22]. In Minnesota, the accuracies for several major crop types are close to or above 95% [23]. Therefore, a result of visual interpretation of multi-temporal Sentinel-2 images based on CDL data was used to select the crop samples for training and testing our crop classification model.

Based on the CDL, crop types in the study area were classified as corn, soybeans, sugar beets, spring wheat, and "other." The latter category ("other") includes all surface cover types except for the four major crops. To ensure the representativeness of the samples and the data size requirements of the deep learning model, the samples are selected to ensure that the sample points are distributed throughout the study area, that the central sample pixel type is consistent with the type of surrounding pixels, and that the sample pixel type is the dominant type in the local neighborhood. The sample points were created from a function of randomly created points and labeled by visual interpretation. Table 3 details the samples used for training the classification model and evaluating the accuracy. To train the model, the training and validation samples in Table 3 are randomly divided into training samples and validation samples in a ratio of 7:3.

**Table 3.** The five categories used in the present study for classification and the number of samples.

| Sample Type | Training and Validation Samples | Testing Samples |
|---|---|---|
| Corn | 1481 | 4096 |
| Soybeans | 1487 | 4738 |
| Spring Wheat | 1445 | 4674 |
| Sugarbeets | 1471 | 4167 |
| Others | 1546 | 5210 |

## 3. Methodology

### 3.1. Methodological Overview

The overall workflow of this study is shown in Figure 2. Firstly, we selected samples as described in Section 2.2.2. Next, different time-series images were constructed for the subsequent classification experiments (Section 3.2). Multiple deep learning models were constructed (Section 3.5), in which random forest was used as benchmark model. Details of the experiments can be found in Section 3.6. Finally, all classification results were validated, compared and analyzed.

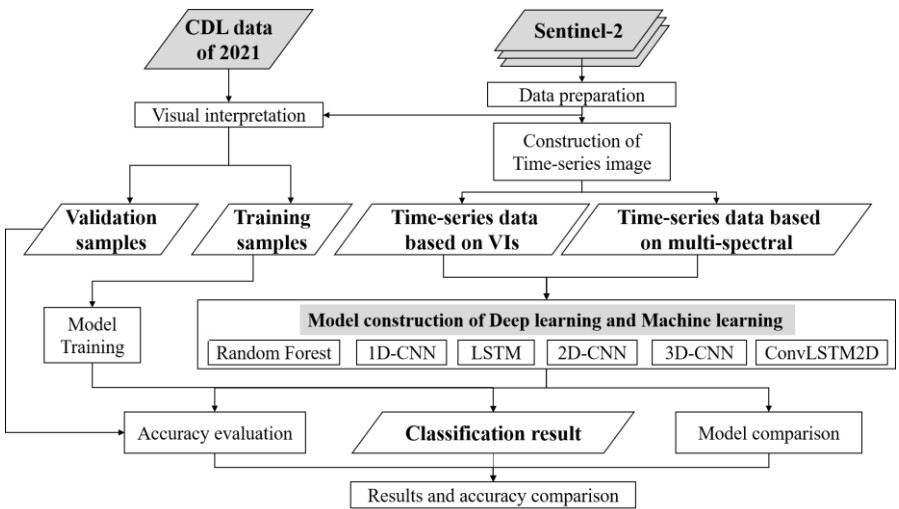

**Figure 2.** General workflow of this study.

### 3.2. Temporal Phenological Patterns

Two main strategies are available to represent the temporal patterns of crops for multi-temporal image crop classification: (1) time-series VIs constructed from spectral characteristics, and (2) time-series multi-spectral bands based on spectral stacking [5], which means stacking multi-temporal images by time sequence. Both strategies have been used to construct time-series data to represent the temporal characteristics of crops. Given the sensitivity of the NDVI [24] and EVI [25] to the physiological state of vegetation and their wide application [5,9], these indices have been used to construct time-series data. Their formulas are as follows:

$$NDVI = (NIR - RED)/(NIR + RED) \tag{1}$$

$$EVI = G \times (NIR - RED)/(NIR + C_1 \times RED - C_2 \times BLUE + L), \tag{2}$$

where $G = 2.5$, $C_1 = 6.0$, $C_2 = 7.5$, and $L = 1.0$. *NIR*, *RED* and *BLUE* represent the spectral reflectance bands of B8(NIR), B4(Red) and B2(Blue) in Sentinel-2 (Table 1).

### 3.3. Deep Learning Models

A CNN is a multilayer feed-forward neural network. The advantages of local connectivity and weight sharing not only decrease the number of parameters but also reduce the complexity of the model and make CNNs more suitable for processing numerous images [9,26]. CNNs may be one-dimensional (1D-CNN), two-dimensional (2D-CNN), or three-dimensional (3D-CNN), by having convolution kernels of different dimensions. Sequence data are fed into 1D-CNNs for learning and representing sequence relationships. Patch-based 2D-CNNs can be used for learning and representing spatial and spectral features in images. Cube-based 3D-CNNs correspond to the spectral, spatial, and temporal features in multi-temporal images [12,14]. The LSTM solves the problems of vanishing gradient, exploding gradient, and deficiencies in long-term dependency representation

that appear in RNNs. In LSTM, the gate mechanisms, which include the input gate, output gate, and forget gate, enhance or weaken the state of the data in the cell for information protection and control [16,17]. The ConvLSTM model is an improvement and extension of the LSTM model, wherein matrix multiplication in LSTM is replaced by a convolution at each gate [20]. The ConvLSTM model combines the structural advantages of LSTM and CNN, and not only captures the spatial context of the image, but also models the long-term dependencies in the spectral domain. In addition, inter- and intra-layer data transfer enables the ConvLSTM to extract features more efficiently than a CNN or LSTM [18,19].

*3.4. Sample Dimensions*

Limited by the size and dimensions of samples in multi-temporal RS images, classification samples contain different spectral, temporal, and spatial information. This study uses various deep learning models to learn and represent spectral, temporal, and spatial information from multi-temporal images. The time-series classification data constructed from VI have only temporal characteristics [9], and their samples are one-dimensional vectors (Figure 3a). The time-series data constructed directly using multi-spectral, multi-temporal images are two-dimensional matrices with the shape of (band, time) (Figure 3b). The time-series data constructed from VIs including the spatial neighborhood are three-dimensional matrices (Figure 3c) with the shape of (height, width, time). The multi-spectral features combined with the spatial neighborhood in multi-temporal images produce four-dimensional matrices with the shape of (time, height, width, band) (Figure 3d). The "time" in three- or four-dimensional matrices means the number of temporals in time-series.

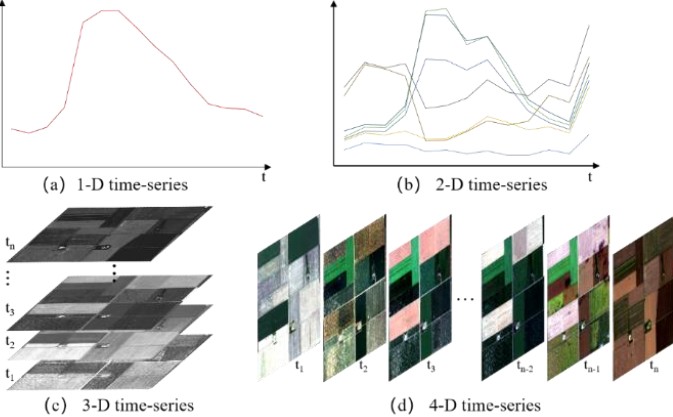

**Figure 3.** Time-series samples with different dimensions. (**a**) 1-D time-series, (**b**) 2-D time-series, (**c**) 3-D time-series, (**d**) 4-D time-series.

*3.5. Deep Learning Architectures*

The main deep learning classification models used in the study are 1D-CNN, LSTM, 2D-CNN, 3D-CNN, and ConvLSTM2D. The temporal, spectral, and spatial information of multi-temporal images can be learned and represented by different deep learning models corresponding to different types of samples. Both 1D-CNN and LSTM models can represent temporal features, and the model input corresponds to 1D and 2D samples (Figure 3a,b). 1D-CNN (Conv1D) models acquire the temporal patterns of sequence data through a 1D convolution, and Conv1D layers learn local features by stacking in a shallow network, whereas a deeper network synthesizes more pattern features within a larger receptive field. The representation of sequence patterns by LSTM models at different temporal frequencies is advantageous for analyzing the temporal characteristics within a crop growing season. 3D times-series samples (Figure 3c) are used as 2D-CNN input, and the Conv2D layer captures the crop temporal and spatial variations through convolution of the spatial domain and through time sequences of the multi-temporal images. 3D-CNN convolves multi-temporal images from different dimensions and represents features of shallow and deep temporal, spatial, and spectral information of crops by stacking convolutional layers (Conv3D). Like

LSTM, ConvLSTM2D is sensitive to temporal patterns, and convolutional operations inside the ConvLSTM2D cell efficiently capture spatial information. The structure (ConvLSTM2D) learns and represents temporal, spectral, and spatial information similar to that of the 3D-CNN models. Both 3D-CNN and ConvLSTM2D models use 4D time-series samples (Figure 3d) as model input. Figure 4 shows the different network architectures.

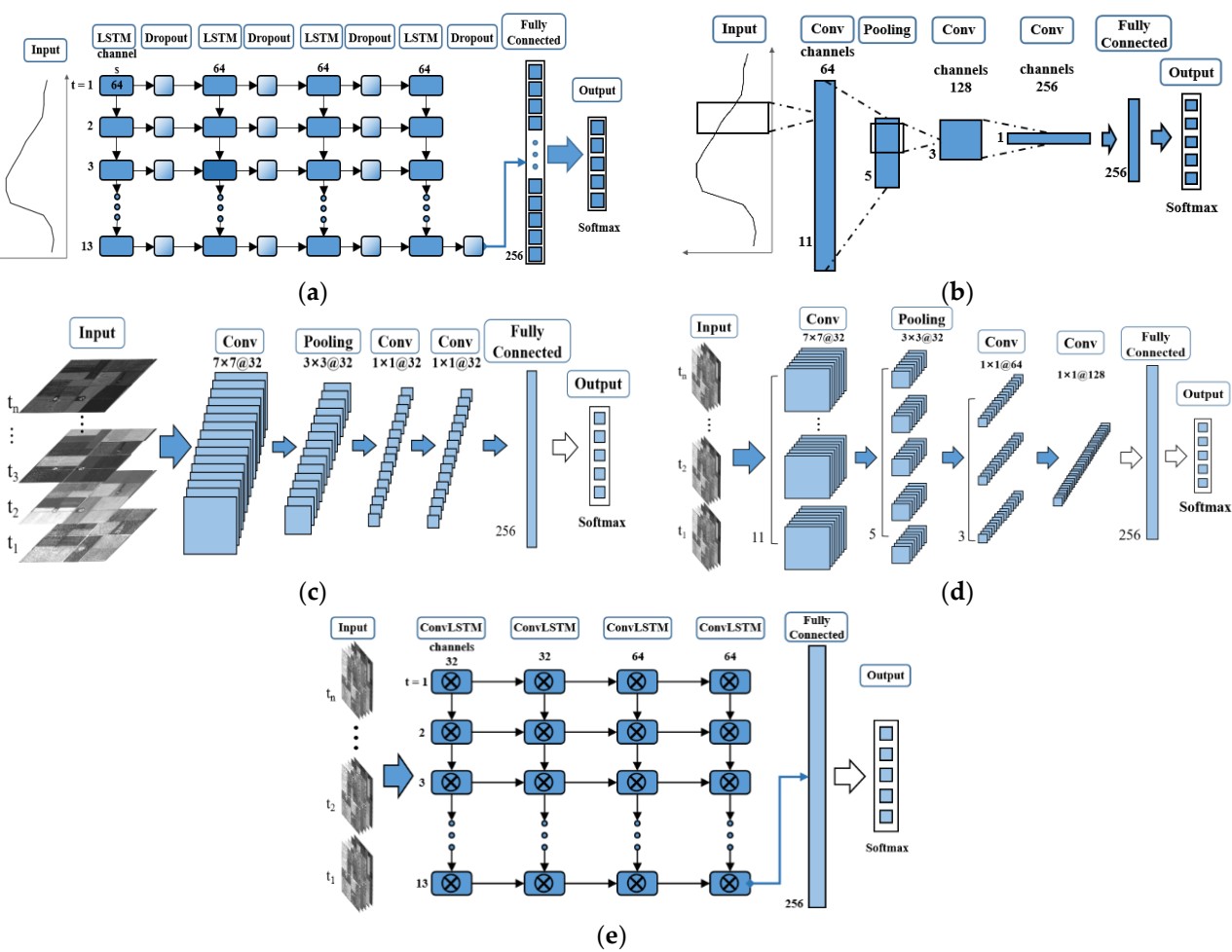

**Figure 4.** Architectures of (**a**) LSTM, (**b**) 1D-CNN, (**c**) 2D-CNN, (**d**) 3D-CNN, and (**e**) ConvLSTM2D.

Because of the versatility and complexity of deep learning architectures, no standard procedure exists to search for the optimal combination of hyperparameters and the associated layers [18,19]. As a result, an extremely large number of potential network architectures must be considered, making it impossible to try them all. In this paper, the hyperparameter setting and optimization of model are based on strategies from the literature [8,9]. The hyperparameters of the deep learning models include the type and number of hidden layers and the number of neurons in each layer. The layer channels are 16, 32, 64, 128, 256 and the sample sizes are 3, 5, 7, 9. The learning rate is 0.01 or 0.05. The length of the time series is 13. The convolution kernel width is 3 [26,27]. Pooling layers are fixed as max-pooling, with a window size of 2. Dropout with probabilities of 0.3, 0.5, and 0.8 is a regularization technique that randomly drops neurons in a layer during training to prevent the output of the layer from relying on only a few neurons. Each model contains two fully connected layers at the output end. The last layer contains five neurons corresponding to the probability of the five classes.

The hyper-parameters are selected and determined step-by-step, based on numerous training experiments. Each deep learning model (Figure 4) is determined by stepwise optimization and adjustment [9]. A large number of training experiments have shown

that the epoch of 400 can meet the training requirements of the model. All deep learning architectures are trained by a backpropagation algorithm, where the stochastic gradient descent is used as the optimizer for model training. The parameters of the stochastic gradient descent are decay $= 10 - 5$ and momentum $= 0.99$. The sample size of the architectures is 9. The learning rate and batch size are 0.01 and 32, respectively. The dropout probability in LSTM is 0.8. Binary cross entropy serves as the loss function. Deep learning models were built using the Keras library and TensorFlow. Finally, the confusion matrix and kappa coefficient from Scikit-learn are metrics for evaluating the accuracy of crop classification. The calculation of VIs and the construction of time-series data are implemented in Python.

### 3.6. Experiment Design

The multi-temporal images are divided into different experimental groups based on the multiple sample types presented in Section 3.2, and the different deep learning models are used to classify the crops based on multi-temporal images. Additionally, random forest is used as a benchmark model in E1, E2, E3 and E6. See Table 4 for details. The B2348 (Table 4) corresponds to the four spectral bands in Table 1. The same applies to the other features (Table 4).

**Table 4.** Experiment groups.

| Number | Features | Samples Dimensions | Model | |
|--------|----------|--------------------|-------|---|
| E1 | NDVI | 1-D time-series | 1D-CNN | LSTM |
| E2 | EVI | | | |
| E3 | B2348 | 2-D time-series | 1D-CNN | LSTM |
| E4 | B2348 + B11 + B12 | | | |
| E5 | B2345678 | | | |
| E6 | All Bands | | | |
| E7 | NDVI | 3-D time-series | 2D-CNN | |
| E8 | EVI | | | |
| E9 | B2348 | 4-D time-series | 3D-CNN | ConvLSTM-2D |
| E10 | All Bands | | | |

E1 and E2 are time-series VI datasets with temporal features constructed from a single VI. E3–E6 are multi-temporal images acquired with different spectral combinations. E3 is a conventional spectral combination of red–green–blue and near-infrared bands. E4 and E5 add shortwave infrared (SWIR) and red-edge spectral bands to E3, respectively. E6 contains the 10 spectral bands of Sentinel-2 images. 1D-CNN and LSTM models are used for crop classification with different spectral combinations and to analyze how multi-spectral and temporal information affect classification accuracy. E7 and E8 are used to classify crops with a 2D-CNN model, and the comparison with E1 and E2 is designed to quantify the contribution of spatial information in multi-temporal crop classification. E9 and E10 are used to classify crops with 3D-CNN and ConvLSTM2D models; E9 uses conventional spectral bands as input and E10 uses the 10 spectral bands of Sentinel-2 images. The comparison and analysis of crop classification with the different experimental groups show how temporal, spectral, and spatial information affect classification accuracy.

### 4. Results

The accuracy of crop classification via multi-temporal images mainly depends on three factors: time-series data construction, feature extraction, and classification method. Our experiments verify the contribution of time-series data and deep learning models. Various time-series data are constructed based on the strategy presented in Section 3.2 and feed

into the deep learning architectures (Figure 4) of Section 3.3 for different experiments. The classification results and accuracies are given in subsequent sections.

### 4.1. Classification Based on VI Time Series

E1 and E2 in Figure 5 and Table 5 show the results of time-series crop classification based on NDVI and EVI. The classification accuracies produced by the 1D-CNN (Figure 4b) and LSTM (Figure 4a) models for E1 and E2 exceed 92%, and the kappa coefficient is greater than 0.9. The highest overall accuracy (OA) for E2 (LSTM) is close to 94%. Compared with random forest, deep learning models based on 1D-CNN and LSTM have higher accuracy (Table 5) and better performance in local regions (Figure 5). These results show that the 1D-CNN and LSTM models constructed herein are suitable for multi-temporal crop classification based on VI. Compared with E1, the OA for E2 increases by 0.26% and 0.69% for the 1D-CNN and LSTM models, respectively. This reflects the variability of different VIs and the similarity of time-series VI for crop classification. Compared with the 1D-CNN model, the LSTM model is more accurate; the OA improves by 0.75% and 1.18% for E1 and E2, respectively. These results show that both the LSTM and 1D-CNN models can capture temporal features, although the LSTM model is more accurate.

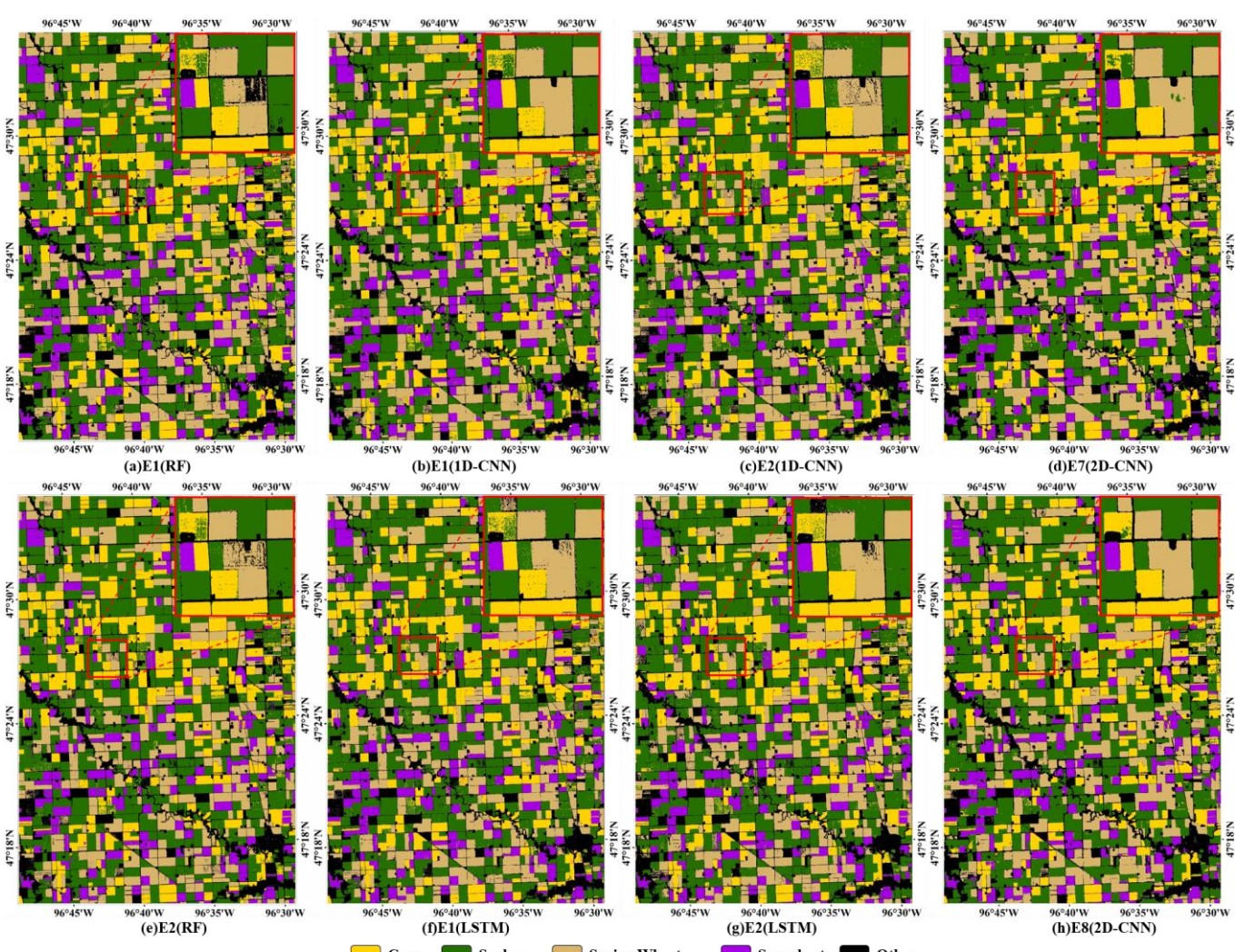

**Figure 5.** Crop classification results based on VI time-series (see red boxes for more detail).

**Table 5.** Classification accuracy produced by various models with VI time series.

| Number | Model | Accuracy | |
|---|---|---|---|
| | | **OA** | **Kappa** |
| | RF | 91.02 | 0.891 |
| E1 | 1D-CNN | 92.50 | 0.906 |
| | LSTM | 93.25 | 0.915 |
| | RF | 91.24 | 0.893 |
| E2 | 1D-CNN | 92.76 | 0.909 |
| | LSTM | 93.94 | 0.924 |
| E7 | 2D-CNN | 94.74 | 0.934 |
| E8 | 2D-CNN | 94.76 | 0.934 |

Differences in architecture also affect classification accuracy. Compared with the other results in Figure 5, the RF-based results (Figure 5a,e) are worse locally, while almost no salt-and-pepper noises appear in Figure 5c,h. Compared with E1 and E2, the accuracy of E7 and E8 improved by 0.82% to 2.24%, and the improvement exceeds RF by 3.5%. E7 and E8 classified by the 2D-CNN model (Figure 4c) produce a favorable overall classification accuracy of above 94.7% and a kappa coefficient of 0.934, which is attributed to the effective learning and representation of temporal and spatial information in patch-based time-series VI data by 2D-CNN.

Figure 5 and Table 5 also show that the classification results based on deep learning outperform the random forest. However, the misclassification of crop types in Figure 5 indicates that further optimization is still needed. Based on the same model, there is no significant accuracy difference in E1 and E2. This indicates that improving accuracy solely using time-series data (temporal features) constructed from a single VI is difficult. However, the addition of spatial information not only improves crop classification accuracy but also eliminates salt-and-pepper noise. In addition, the 1D-CNN and LSTM architectures limit the possibility of exploiting spatial information in multi-temporal crop classification, whereas the 2D-CNN model produces more accurate crop classification based on single VI time-series data.

### 4.2. Classification Based on Multi-Spectral Time Series

Figure 6 and Table 6 show the classification results of E3–E6 based on the time-series data constructed from multi-spectral, multi-temporal images. The crop classification accuracy of the 1D-CNN model is less than that of the LSTM model applied to E3–E6, which is similar to the results of the LSTM model. Therefore, hereinafter, we consider only the crop classification results based on LSTM.

The input data in E3–E6 have both multi-spectral and -temporal features, differing only in the number of multi-spectral bands, as explained in Section 3.4. Table 6 shows that the accuracy of RF-based is lower than deep learning, and Figure 6 also shows that results of deep learning are better in local areas. The OA of E3–E6 is 95.31%, 96.72%, 96.37%, and 96.94%, respectively. Compared with E3, the addition of spectral bands, especially red-edge bands (E5) or SWIR bands (E4), improves the crop classification accuracy, with SWIR bands contributing slightly more than red-edge bands. Using the LSTM model with E6 surprisingly remains the most accurate configuration, with the crop-classification accuracy improving by 1.63% with respect to E3. This indicates that the advantage of the number of spectral bands in multi-spectral images cannot be neglected. With the addition of spectral bands, salt-and-pepper noise is eliminated to varying degrees, with the least salt-and-pepper noise coinciding with the most accurate crop classification (Figure 6f), indicating that the salt-and-pepper phenomenon is weakened but hardly eliminated by using multi-spectral bands. Combined with the presentation in Section 4.1, these results further demonstrate how spatial information affects multi-temporal crop classification.

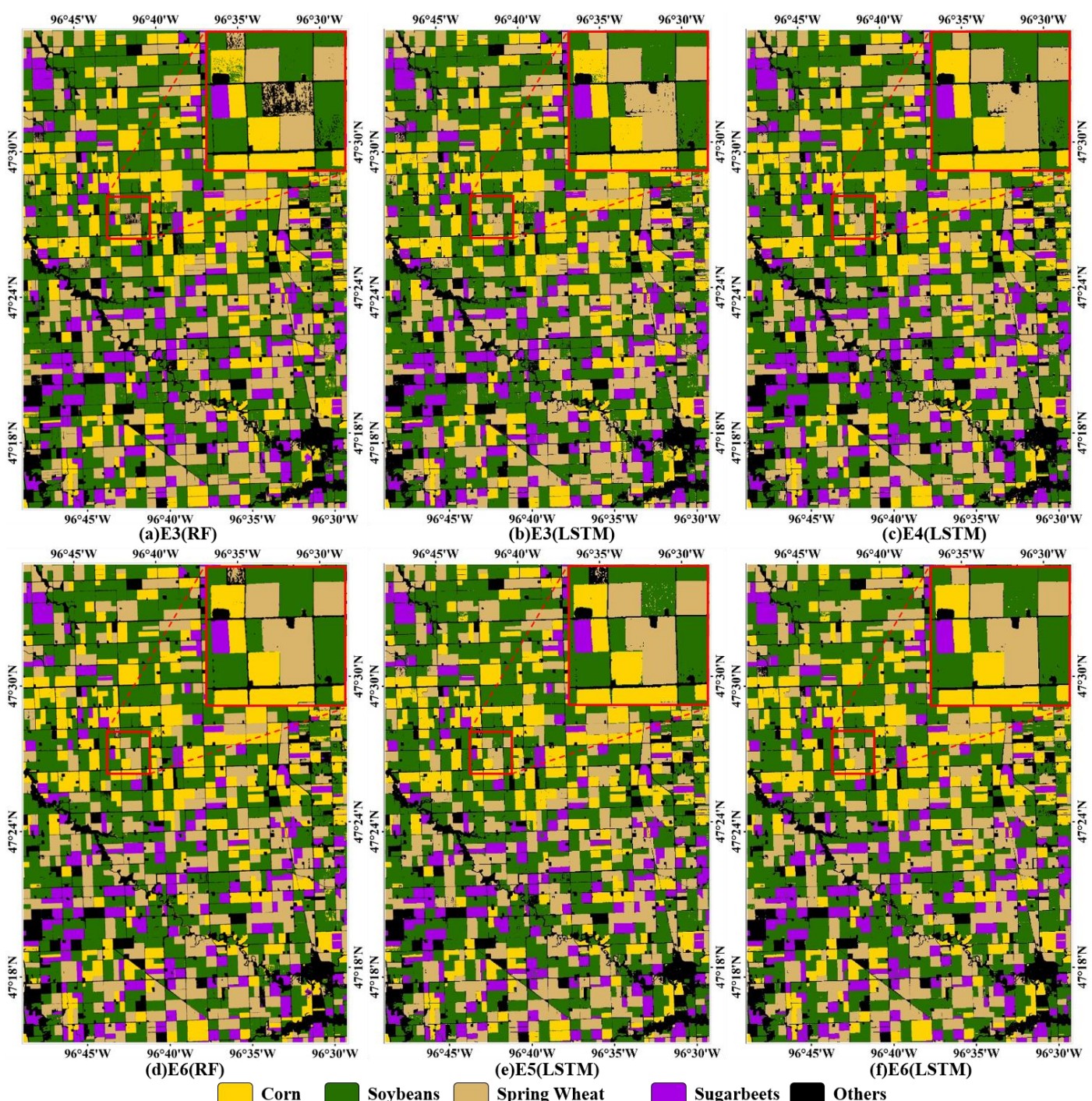

**Figure 6.** Crop classification results based on multi-spectral time series.

Furthermore, the addition of different spectral bands in E3–E6 increases the diversity of input classification data. In the same experimental group, the accuracy difference between 1D-CNN and LSTM varies from 0.1% to 0.44%, with the minimum difference of 0.1% presented in E6. However, in the different experimental groups, the accuracy difference of the same model varies from 1.06% to 1.95%, with E6 showing an accuracy improvement of nearly 2% compared to E3. In E9 and E3, the spatial information causes differences in the input data. The accuracy difference between different deep learning models with the same input data is small, ranging from 0.21% to 0.42%. In contrast, the accuracy difference between the same models with different input data is larger, ranging from 1.88% to 1.25%.

This indicates that increasing the diversity of input data is more important for improving crop classification accuracy than using different deep learning models.

**Table 6.** Classification accuracy produced by various models and multi-spectral time-series data.

| Number | Model | Accuracy | |
|---|---|---|---|
| | | OA | Kappa |
| | RF | 93.48 | 0.918 |
| E3 | 1D-CNN | 94.89 | 0.936 |
| | LSTM | 95.31 | 0.941 |
| E4 | 1D-CNN | 96.28 | 0.953 |
| | LSTM | 96.72 | 0.959 |
| E5 | 1D-CNN | 96.02 | 0.950 |
| | LSTM | 96.37 | 0.955 |
| | RF | 95.51 | 0.944 |
| E6 | 1D-CNN | 96.84 | 0.960 |
| | LSTM | 96.94 | 0.962 |
| E9 | 3D-CNN | 96.77 | 0.960 |
| | ConvLSTM2D | 96.56 | 0.957 |
| E10 | 3D-CNN | **97.43** | **0.968** |
| | ConvLSTM2D | 97.25 | 0.966 |

Figure 7 and Table 6 present the classification results of E9 and E10 using the 3D-CNN (Figure 4d) and ConvLSTM2D (Figure 4e) models. The OA of 3D-CNN in E9 and E10 was 96.77% and 96.56%, respectively, with kappa coefficients of 0.960 and 0.957. The OA of ConvLSTM2D in E9 and E10 was 97.43% and 97.25%, respectively, with kappa coefficients of 0.968 and 0.966. The accuracy is slightly greater when using the 3D-CNN model than when using the ConvLSTM2D model. The use of the 3D-CNN model on E10 produces the greatest crop classification accuracy of 97.43%, which translates into an OA improved by 3.69%, 2.67%, 0.49%, and 4.93% with respect to E2 (LSTM), E8 (2D-CNN), E6 (LSTM), and E1 (1D-CNN), respectively. Compared with the E6 (LSTM), the salt-and-pepper noise is eliminated in E9 and E10 (Figure 7b,d), although the improvement in accuracy is not obvious. E10 produces more accurate results than E9 because it contains more spectral bands in the input data.

The classification results of the different experiments verify the feasibility of the model constructed herein (Figure 4) for multi-temporal crop classification. The comparison of the results of the different experiments shows that both the construction of the time-series data and that of the classification model influence the crop classification accuracy. The LSTM model produces more accurate crop classification results than the 1D-CNN model. However, when using time-series data constructed from VIs, the 2D-CNN model produces more accurate results than the 1D-CNN and LSTM models after the elimination of the salt-and-pepper noise. When using time-series data constructed by stacking spectral bands, increasing the number of bands in the input data improves the crop classification accuracy while somewhat reducing the salt-and-pepper noise. Additionally, the LSTM model again produces slightly more accurate crop classifications than the 1D-CNN model, which indicates that the LSTM model is more able to capture temporal features.

E10 treated by the 3D-CNN and ConvLSTM2D models (Figure 4) produces the most accurate crop classification of all experiments. In addition, the architectures of the 3D-CNN and ConvLSTM2D models lead to better learning and representation for multi-temporal crop features, making these models more suitable for crop classification from multi-temporal images.

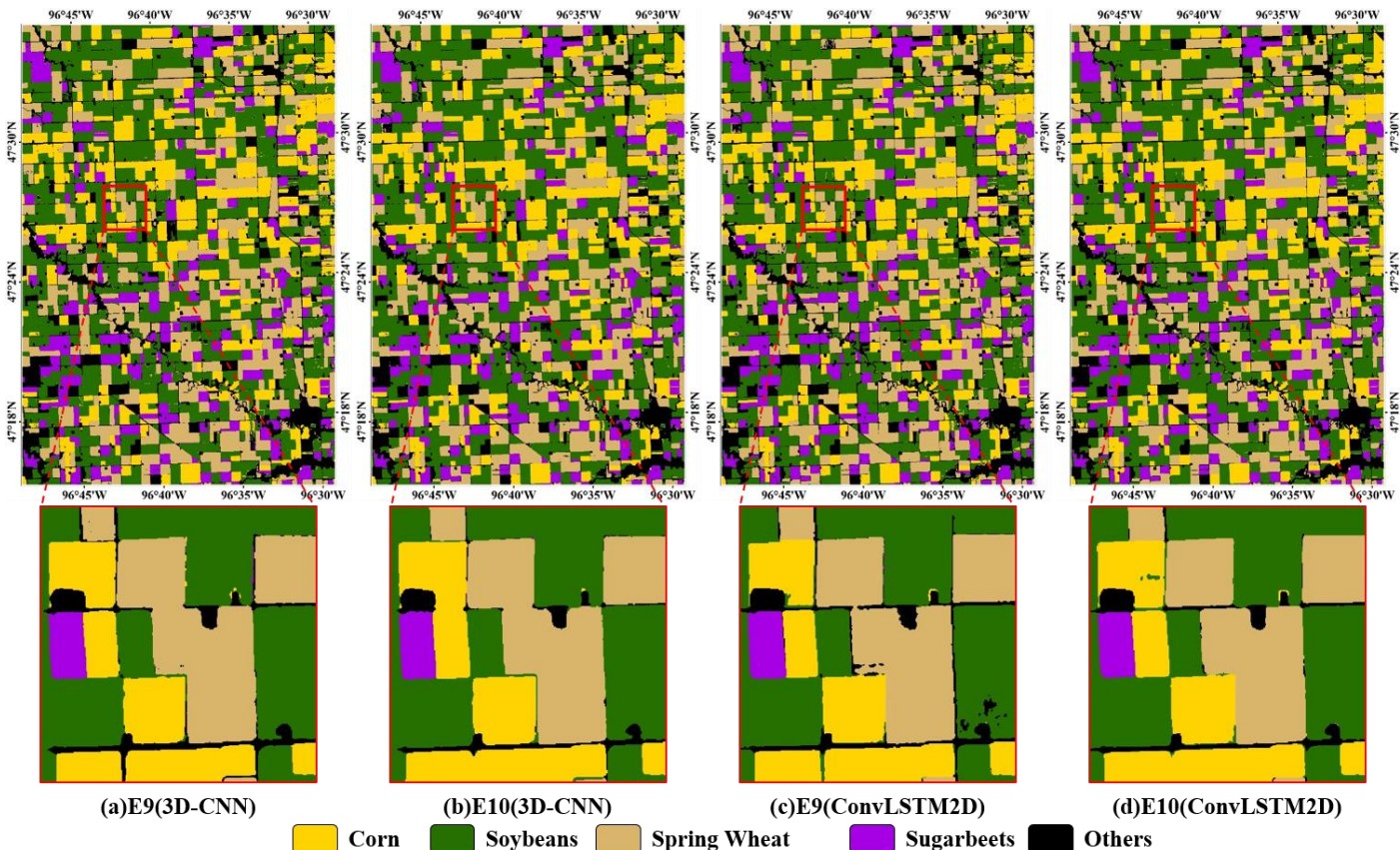

**(a)E9(3D-CNN)**     **(b)E10(3D-CNN)**     **(c)E9(ConvLSTM2D)**     **(d)E10(ConvLSTM2D)**

| Corn | Soybeans | Spring Wheat | Sugarbeets | Others |

**Figure 7.** Crop classification results based on temporal, spectral, and spatial information.

Combined with the previous analysis of classification accuracy, VI time-series data using only temporal information only slightly improves the crop classification accuracy. The addition of multi-spectral data based on temporal information improves crop classification accuracy, and the salt-and-pepper noise is more easily alleviated upon increasing the number of spectral bands. As the number of input features increases, the contribution of spatial information in improving classification accuracy decreases. However, the elimination of salt-and-pepper noise through the use of spatial information remains a clear advantage in crop mapping. Therefore, making full use of the temporal, spectral, and spatial information is a more feasible strategy for multi-temporal crop classification. The deep learning architecture fed with 4D data involving multi-temporal images is thus the best model for accurate crop classification based on multi-temporal images.

## 5. Discussion

### 5.1. Analysis of Time-Series Profile

Figure 8 shows the temporal profiles of crops produced by VIs and spectra. The buffer areas of crop profiles overlap throughout the growing season, despite the difference in average reflectance or VI values. In the middle of the growing season, the spectral overlap within the crop becomes smaller (~DOY 200–220) than in the early or late growing season. During this period, the temporal curves of crops with one standard deviation are more stable and distinguishable, which indicates that this feature should be useful for differentiating between crops. In addition, the temporal windows always serve for single-temporal crop classification [28]. However, the similarity and overlap of profiles over the whole growing season make it difficult to distinguish crops such as corn and soybeans based solely on single images [1,2].

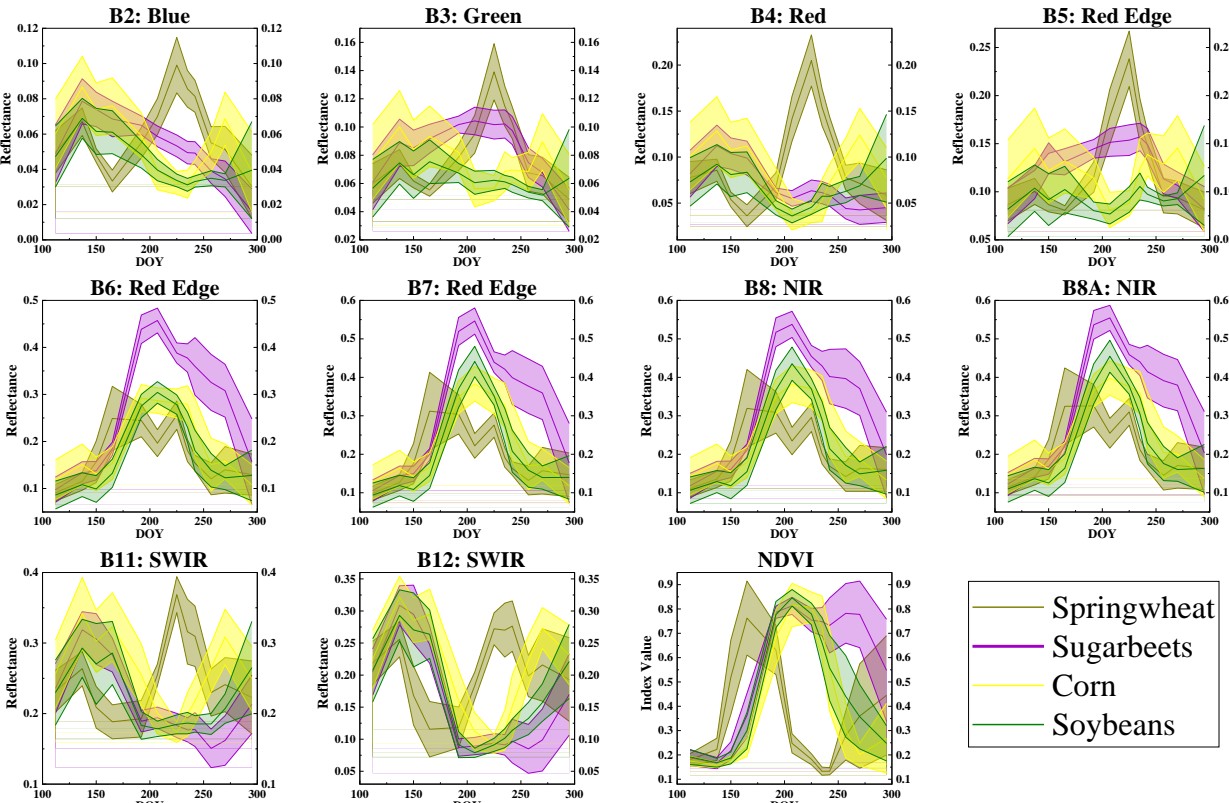

**Figure 8.** Time-series spectral band and vegetation indices are aggregated for crop fields. The buffers indicate one standard deviation calculated from the fields.

The differences in the time series curves (Figure 8) between crops in different spectral ranges and time periods make it possible to distinguish between crops [29]. For example, the gap in B8 (Figure 8) during the middle growing season (≈DOY 180–220) makes it possible to distinguish between spring wheat and sugar beets. Figure 8 shows that almost no spectral overlap occurs between corn and soybeans in B11 and B12 during the period of time (≈DOY 170–200). The gap observed in the profiles of sugarbeets and other crops in bands B6-B8 and B8A, as shown in Figure 8, occurs during two periods of time, which are around DOY 180–220 and 250–270. Spring wheat can be directly distinguished from profiles in B2–B5 (Figure 8) around DOY 225 and in B11 and B12 in the period DOY 210–240. Corn and soybeans can be differentiated with greater probability in the period DOY 170–200 in B11 and B12. In addition, the overlap in temporal profile based on the NDVI is similar to the other spectra in Figure 8. The profiles of corn and soybeans almost overlap over the entire growing season, which explains the difficulty of distinguishing between these two crops [3,4]. The profiles of sugarbeets and spring wheat clearly differ between DOY 260 and 170.

As previously mentioned, time-series images based on single VI or band are insufficient to accurately distinguish between different crops. However, different crops exhibit spectral differences in the time-series curves of each spectral band (Figure 8), indicating the potential of each spectral band to distinguish between different crops. Better utilization of the advantages of multi-spectral bands has greater potential to improve the accuracy of crop classification [30]. The addition of different types of spectral bands such as red-edge and SWIR has reinforced this conclusion in classification experiments [9].

### 5.2. Effects of Temporal, Spectral, and Spatial Feature

The effects of temporal, spectral, and spatial information on crop classification are revealed in the different time-series data. The crop classification results due to the different time-series classification data are shown in Figures 5–7 and Tables 5 and 6. Using only

temporal features may not be sufficient for accurate crop classification due to salt-and-pepper noise (Figures 5 and 6), which can affect pixel-based classification. Fully exploiting the abundant spectral and spatial information in multi-temporal images can be challenging when using only VI, but it provides more possibilities for improving accuracy. [5,31] pointed out that spatial features such as texture can lead to good classification performance, and a similar result occurs for 2D-CNN classification (Figure 5). In addition, based on the analysis in the previous sections, the contribution to the accuracy of spatial information such as texture [9,32] decreases as the number of input features increases. Moreover, the spatial information contributes significantly to the classification accuracy for a feature input of a single VI. [8] also suggested that more information-dense data are required to improve the crop-classification accuracy based on multi-temporal images. The diversity of information and the differences in time-series data depicted in Figure 8 provide more possibilities for accurate classification and can alleviate the salt-and-pepper phenomenon. Nevertheless, spatial information remains a vital ingredient to eliminate salt-and-pepper noise.

*5.3. Comparison of Deep Learning Models*

The temporal dependencies in multi-temporal images are long term and complex, and crops have unique temporal, spectral, and spatial features (Figure 8). Sufficient model complexity and automated feature learning and representation satisfy the data-processing needs of models in multi-temporal crop classification [9,12]. Differing from the result that 1D-CNN accuracy is higher than that of LSTM [9], increasing the number of spectral bands in this work causes the accuracy of 1D-CNN to be close to that of LSTM. This indicates that input features and application scenarios (more crop types) may also affect the accuracy of the classification. The architecture of 2D-CNN models is limited by their structure, meaning that they can only accept time-series data constructed by a single VI or spectral band as input. This prevents 2D-CNN models from exploiting multi-spectral information. The analysis in Section 4 also points out that 2D-CNN models are less accurate than 1D-CNN and LSTM models using multiple spectral bands. In contrast with 2D-CNN models, both 3D-CNN and ConvLSTM2D models require 4D data that perfectly fit the temporal, spectral, and spatial features. The classification results (Figures 7 and 9) of 3D-CNN and ConvLSTM2D models are also significantly more accurate and stable than other comparative models. [30] also pointed out that models such as 3D-CNN should be considered for crop classification from multi-temporal images.

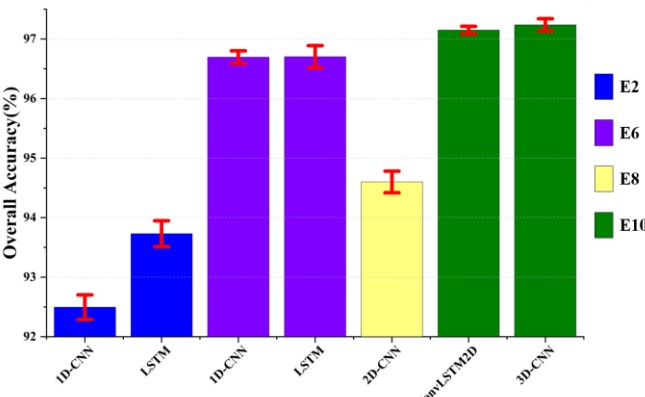

**Figure 9.** The OA of different deep learning models.

As described in Section 3.5, each model is trained extensively to achieve the best classification results. Therefore, the parameters of deep learning models in this work will likely need to be adjusted to achieve satisfactory accuracy for other classification tasks. Additionally, numerous model training experiments are necessary in this process.

### 5.4. Potential of 3D-CNN and ConvLSTM2D for Crop Classification from Multi-Temporal Images

Crop classification from multi-temporal RS images often has a time lag due to data acquisition [5,6]. However, time-series data can alleviate this issue, whereby different objects have the same spectrum, and the same objects have different spectra in the background of relatively complex crop cultivations. Previous analyses also revealed that fully exploiting the temporal, spectral, and spatial information in multi-temporal images should be a major avenue to improve classification accuracy. 3D-CNN and ConvLSTM2D models can integrate multi-temporal information and have advantageous structures not found in other models such as 2D-CNN and SVM [11,19]. The best classification accuracies are provided by 3D-CNN and ConvLSTM2D models, and exceed 97% (Table 6). Figure 10 shows the strong correlation between the results obtained herein and the CDL for the area ratio of different crops. It also shows potential applications for crop classification based on multi-temporal images.

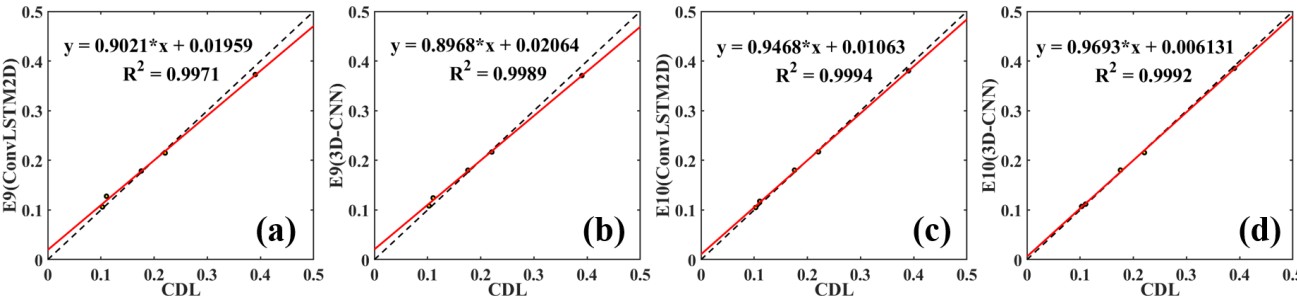

**Figure 10.** Correlation of crop-area ratio. ((**a**–**d**) correspond to four experiments, as shown in the vertical label. The scatter points mean the fraction of different crop over the study area. The red line reflects the consistency of crop area between the classification results and the CDL.)

Different network structures in deep learning models such as inception [33], dropout [8], and transformer [34] all enhance the feature learning and representation capabilities of the network. Deep learning models (Figure 4) are constructed by simple stacking of modules, so they lack special design for multi-temporal images and cannot treat scale effects [35] in images. In addition, information redundancies (Figure 8) with high inter-band similarity must be considered. Both architectures have inherent advantages for processing multi-temporal images. Although ConvLSTM2D has fewer applications in multi-temporal image crop classification than 3D-CNN [14], the results of this study show that this model approaches the classification capability of 3D-CNN. References [13,36] pointed out that 3D-CNN is not suitable for establishing long-term dependencies of time-series data due to locally computed convolutions, whereas ConvLSTM2D combines the sequence processing capability of LSTM and the structure of CNN, which facilitates the addition of multiple special structures and modules so that it can be exploited to classify crops from multi-temporal images.

### 6. Conclusions

This paper constructs various time-series datasets based on Sentinel-2 multi-temporal images by VI or spectral stacking, and develops deep learning models with different structures for classifying crops from multi-temporal images. The results lead to the following conclusions:

(1) Greater data diversity (temporal, spectral and spatial information) is effective in improving crop classification accuracy. The temporal feature only provides limited improvement in the accuracy of crop classification from multi-temporal images. As more spectral information is added, the accuracy can be further improved, and the impact of salt-and-pepper noise can be alleviated. The inclusion of spatial information can eliminate salt-and-pepper noise, and its contribution to accuracy decreases as the number of input features increases.

(2) Various deep learning models have limitations in crop classification from multitemporal images. 1D-CNN and LSTM models cannot extract spatial features while integrating temporal and spectral features. Additionally, a 2D-CNN is suitable for crop classification of time-series data given a single feature such as a VI or band because the multi-spectral advantages are hard to consider when combining temporal and spatial information. The 3D-CNN and ConvLSTM2D models are the most accurate for classifying crops and are more suitable for multi-temporal crop classification than other deep learning models.

(3) The deep learning models based on Conv3D and ConvLSTM2D, which integrate temporal, spectral, and spatial information, are the most accurate models for multitemporal crop classification. In addition, the advantages of incorporating RNN and CNN and the more flexible structure mean that ConvLSTM should be investigated.

In this paper, smaller areas and simple crop types are used for deep learning multitemporal crop classification application studies. In future research, crop classification based on deep learning is still needed for large-scale study areas and complex planting systems, such as crop rotation and more crop types. In addition, the impact of clouds on image acquisition is difficult to avoid. While the acquisition of synthetic aperture radar (SAR) is not affected by clouds, which can also increase the diversity of classification data. Therefore, research into crop classification by synergistic SAR and optical images with different acquisition frequencies will be carried out. Additionally, the ConvLSTM model will be used as the classification model to explore its potential in multi-source image crop classification.

**Author Contributions:** Conceptualization, Q.L. and J.T.; Methodology, Q.L. and J.T.; Software, Q.L.; Validation, Q.L.; Formal analysis, Q.L. and J.T.; Writing—original draft preparation, Q.L.; Writing—review and editing, Q.L., J.T. and Q.T.; Visualization, Q.L.; Project administration, Q.T.; Funding acquisition, J.T. All authors have read and agreed to the published version of the manuscript.

**Funding:** This research was funded by the National Natural Science Foundation of China (grant number: 42101321 and 41771370) and the Open Fund of State Key Laboratory of Remote Sensing Science (grant number: OFSLRSS202119).

**Institutional Review Board Statement:** Not applicable.

**Data Availability Statement:** The data presented in this study are available on request from the author.

**Acknowledgments:** The authors acknowledge the support provided by their respective institutions.

**Conflicts of Interest:** The authors declare no conflict of interest.

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
