# Peer review of "Deep Learning Application for Crop Classification via Multi-Temporal Remote Sensing Images"

_agriculture, doi:10.3390/agriculture13040906_

Round 1

Reviewer 1 Report (New Reviewer)

In this paper, some classifiers especially deep learning approaches are used to classify crops from multi-temporal remote sensing images.

-          Title: it seems that a paper regarding applications of deep learning for crop classification is presented, but it is a general title.

-          Lines 22 to 24: it is possible to remove these sentences.

-          Line 25 and 26: please revise this sentence.

-          Lines 28 and 29: please revise this sentence.

-          Line 68: please revise this sentence.

-          It is necessary to revise the manuscript by a native. There are some errors and mistakes.

-          Figure 1: please insert a false color image based on Sentinel2’ bands.

-          Section 2.2.3 must be presented in the methodology section.

-          Please start the methodology section by a workflow and a paragraph regarding descriptions of the method. Then, present subsections such as preprocessing, spectral features, deep learning approaches.

-          Equations 1, 2: please define all variables.

-          Table 3: please define all variables, B2348?

-          It is interesting for readers to know values selected for parameters of the classifiers. Change of the values may affect on final results, so it should be mentioned in the limitations of the study.

-          In the conclusion section, please present future works for readers.

Author Response

 Response to Reviewer Comments

Overall comments:

In this paper, some classifiers especially deep learning approaches are used to classify crops from multi-temporal remote sensing images.

Response: Thank you so much for your comments.

Your suggestions and comments are very helpful to improve our manuscript. We have revised the manuscript accordingly. Please find our response below.

Point 1: Lines 22 to 24: it is possible to remove these sentences.

Response 1: Thank you so much for your suggestions. We have removed these sentences.

Point 2: Line 25 and 26: please revise this sentence.

Response 2: Thank you so much for your comment. We have revised it.

Line 22: The results show that the accuracy of both 1D-CNN (92.5%) and LSTM (93.25%) is higher than that of random forest (~ 91%) when using a single temporal feature as input.

Point 3: Lines 28 and 29: please revise this sentence..

Response 13: Thank you so much for your comment. We have revised it.

Line 25: The accuracy of 1D-CNN and LSTM models integrated with temporal and multi-spectral features is 96.94% and 96.84%, respectively.

Point 4: Line 68: please revise this sentence.

Response 4: Thank you so much for your comment. We have revised it.

Line 65: Current multi-temporal RS images are multi-spectral, multi-temporal and multi-spatial.

Point 5: It is necessary to revise the manuscript by a native. There are some errors and mistakes.

Response 5: Thank you so much for your comment. We regret there were problems with the language. The paper has been carefully revised by a professional language editing service to improve the grammar and readability. We carefully checked and revised it again.

Point 6: Figure 1: please insert a false color image based on Sentinel2’ bands.

Response 6: Thank you so much for your comment. We have added the false color image in Figure 1.

Figure 1. False color image and The Cropland Data Layer (CDL) of study areas.

Point 7: Section 2.2.3 must be presented in the methodology section.

Response 7: Thank you so much for your comment. We revised the manuscript.

Point 8: Please start the methodology section by a workflow and a paragraph regarding descriptions of the method. Then, present subsections such as preprocessing, spectral features, deep learning approaches.

Response 8: Thank you so much for your comment. We have added new section (3.1 Methodological overview). Figure 2 and a brief description present a clear view.

Figure 2. General workflow of this study

Line 162: The overall workflow of this study is shown in Figure 2. Firstly, we selected sam-ples as described in Section 2.2.2. Next, different time-series images were constructed for the subsequent classification experiments (Section 3.2). Multiple deep learning models were constructed (Section 3.5) in which random forest was used as benchmark model. Details of the experiments can be found in Section 3.6. Finally, all classification results were validated, compared and analyzed.

Point 9: Equations 1, 2: please define all variables.

Response 9: Thank you so much for your comment. We have revised it.

Line 181: NIR, RED and BLUE represent the spectral reflectance bands of B8(NIR), B4(Red) and B2(Blue) in Sentinel-2 (Table 1).

Point 9: Table 3: please define all variables, B2348?.

Response 9: Thank you so much for your comment. We have added Table 1 to describe the spectral bands. We have also added descriptions of other variables.

Line 277: The B2348(Table 4) corresponds to the four spectral bands in Table 1. The same applies to other Features (Table 4).

Table 1. Spectral bands of Sentinel-2 images.

Band Names

Spectral Band

Central Wavelength(nm)

Band Names

Spectral Band

Central Wavelength(nm)

Blue

B2

490

Red-Edge

B7

775

Green

B3

560

NIR

B8

842

Red

B4

665

NIR

B8a

865

Red-Edge

B5

705

SWIR

B11

1610

Red-Edge

B6

740

SWIR

B12

2190

 Point 9: It is interesting for readers to know values selected for parameters of the classifiers. Change of the values may affect on final results, so it should be mentioned in the limitations of the study.

Response 9: Thank you so much for your comment. We have added the description of the selection of parameters.

In section 3.5, we introduce the hyperparameters of the deep learning models and the candidate ranges of parameters. Reference [9] also points out that the selection and determination of the parameters of the network model are determined by extensive training experiments. And this is the common way of determining the parameters of current deep learning networks [8,9].

We agree that change of the parameter values may affect on final results. And the determination of the final network parameters in the manuscript was accomplished by extensive parameter fine-tuning. The best training parameters and model structure parameters in the manuscript are shown in Figure 4 and section 3.5 (Line 261-272).

In addition, we also add the description in section 5.3 that numerous model training experiments are necessary to detemine the optimal deep learning model.

Line 492: As described in Section 3.5 each model is trained extensively to achieve the best classification results. Therefore, the parameters of deep learning models in this work will likely need to be adjusted to achieve satisfactory accuracy for other classification tasks. And numerous model training experiments are necessary in this process.

  1. Dou, P.; Shen, H.; Li, Z.; Guan, X. Time series remote sensing image classification framework using combination of deep learning and multiple classifiers system. International Journal of Applied Earth Observation and Geoinformation. 2021, 103, 102477. https://doi.org/10.1016/j.jag.2021.102477.
  2. Zhong, L.; Hu, L.; Zhou, H. Deep learning based multi-temporal crop classification. Remote Sens. Environ. 2018, 221, 430-443. https://doi.org/10.1016/j.rse.2018.11.032.

Point 10: In the conclusion section, please present future works for readers.

Response 9: Thank you so much for your comment. We have added the description of future works.

Line 548: In this paper, smaller areas and simple crop types are used for deep learning multi-temporal crop classification application studies. In future research, crop classification based on deep learning is still needed for large-scale study areas and complex planting systems, such as crop rotation and more crop types. In addition, the impact of clouds on image acquisition is difficult to avoid. While the acquisition of Synthetic Aperture Radar (SAR) is not affected by clouds, which can also increase the diversity of classification data. Therefore, the research of crop classification by synergistic SAR and optical images with different acquisition frequencies will be carried out. And the ConvLSTM model is used as the classification model to explore its potential in multi-source image crop classification.

Finally, we would like to thank the reviewer again for your comments!

Reviewer 2 Report (New Reviewer)

This paper develops various time-series data sets based on Sentinel-2 multi-temporal images by VI or spectral stacking and develops deep learning models with different structures for classifying crops from multi-temporal images. This work is well-written and timely. Comments to the authors:

1) Provide details about spectral stacking.

2) How the testing and training samples are selected for this work.

3) Provide the detailed implementation in a flowchart for a clear view.

4) Provide suitable references for the equations considered in this work?

5) Implementation of deep learning algorithms is not clear. Provide more details.

6) What are measures used to find the accuracy of the models proposed?

Some minor grammatical and spell checks are required.

Author Response

Response to Reviewer Comments

Overall comments:

This paper develops various time-series data sets based on Sentinel-2 multi-temporal images by VI or spectral stacking and develops deep learning models with different structures for classifying crops from multi-temporal images. This work is well-written and timely.

Response: Thank you so much for your positive comments.

Your suggestions and comments are very helpful to improve our manuscript. We have revised the manuscript accordingly. Please find our response below.

Point 1: Provide details about spectral stacking.

Response 1: Thank you so much for your suggestions. We add the description of spectral stacking.

Line 173: time-series multi-spectral bands based on spectral stacking [5], which means stacking multi-temporal images by time sequence.

Point 2: How the testing and training samples are selected for this work.

Response 2: Thank you so much for your comment.

In Section 2.2.2 we describe the requirements for sample selection and show the details of the samples. In addition, We added the description as a supplement to the sample selection.

Line 154: The sample points were created from function of create random points and labeled by visual interpretation.

Point 3: Provide the detailed implementation in a flowchart for a clear view.

Response 13: Thank you so much for your comment. We have added new section (3.1 Methodological overview). Figure 2 and a brief description present a clear view.

Figure 2. General workflow of this study

Line 162: The overall workflow of this study is shown in Figure 2. Firstly, we selected sam-ples as described in Section 2.2.2. Next, different time-series images were constructed for the subsequent classification experiments (Section 3.2). Multiple deep learning models were constructed (Section 3.5) in which random forest was used as benchmark model. Details of the experiments can be found in Section 3.6. Finally, all classification results were validated, compared and analyzed.

Point 4: Provide suitable references for the equations considered in this work?

Response 4: Thank you so much for your comment. We have added references.

Line 175: Given the sensitivity of the NDVI [24] and EVI [25] to the physiological state of vege-tation and their wide application.

  1. Tucker, C. J. Red and photographic infrared linear combinations for monitoring vegetation. Remote Sens. Environ. 1979. 8(2), 127-150. http://dx.doi.org/10.1016/0034-4257(79)90013-0.
  2. Huete, A., Didan, K., Miura, T., Rodriguez, E., Gao, X., Ferreira, L., Overview of the radiometric and biophysical perfor-mance of the MODIS vegetation indices. Remote Sens. Environ. 2002. 83(1),195-213.http://dx.doi.org/10.1016/S0034-4257(02)00096-2.

Point 5: Implementation of deep learning algorithms is not clear. Provide more details.

Response 5: Thank you so much for your comment. We added and revised the description about the training of deep learning.

In Section 3.5, we clearly describe the structure of the deep learning model (Figure 4) and the selection and determination of the hyperparameters. We also describe the information related to the code libraries used for model building and running.

Point 6: What are measures used to find the accuracy of the models proposed?

Response 6: Thank you so much for your comment.

Deep learning models in this manuscript were constructed for multi-temporal crop classification. And the accuracy of the models proposed is ultimately presented as the accuracy of the classification. Confusion matrix and kappa coefficient were used as metrics for evaluating the accuracy of crop classification. In addition, validation samples were selected to evaluate the accuracy of the models.

Point 7: Some minor grammatical and spell checks are required.

Response 7: Thank you so much for your comment. We regret there were problems with the language. The paper has been carefully revised by a professional language editing service to improve the grammar and readability. We carefully checked and revised it again.

Finally, we would like to thank the reviewer again for your comments!

Round 2

Reviewer 1 Report (New Reviewer)

Accept as it is.

This manuscript is a resubmission of an earlier submission. The following is a list of the peer review reports and author responses from that submission.

Round 1

Reviewer 1 Report

I was hounored honored review the article titled "Deep Learning Application for Crop Classification via Multi-Temporal Remote Sensing Images", authored by Qianjing Li, Jia Tian, and Qingjiu Tian.

This article examines the effectiveness of deep learning models in crop classification using multi-temporal remote sensing images. The authors compared and evaluated the performance of five deep learning architectures: 1D-CNNs, LSTM, 2D-CNNs, 3D-CNNs, and ConvLSTM2D, with different time-series data. The experiments analyzed the effects of temporal, spectral, and spatial information on classification accuracy and explored the most suitable multi-temporal deep learning crop classification strategies. The results indicate that combining temporal features with multi-spectral or spatial information significantly improves classification accuracy. The 3D-CNN and ConvLSTM2D models are the best deep learning architectures for multi-temporal crop classification. However, the authors suggest that the ConvLSTM architecture should be further developed for multi-temporal image crop classification.

In my opinion, the article is fit for publication - after minor editorial corrections. And in detail:

1. While quoting tables and figures - i preffer making them in format as Fig. and Tab. - even in brackets.

2. In table 2 - Please expand the descripcion.

3. In figure 2 - i suggest adding description of a-d. even we may found it in text below.

4. Figure 4 - the legend is truncated - same in fig. 5 and 6.

5. Please provide description in methodology - what software and libraries were used for all calculations.

6. Fig 8 - mayby better quality?

7. line 481 - after bracket - unnecessary dot.

The article is scientific in nature. It is legible and clear. Methods used - properly selected. Results - well presented. Discussion - could be extended - but I consider it sufficient to publish. Conclusions - Right.

Language - proper.

I recommend to the Editors to publish the manuscript with minor changes.

Author Response

Dear Editors and Reviewers,

We would like to thank you and the reviewers for the insightful comments and suggestions to improve this manuscript. We have thoroughly revised the manuscript to address all reviewers’ comments individually. Specifically, we have revised this manuscript on the following major points. (1) We have edited and revised the language of the manuscript. (2) We checked and revised the formatting of the manuscript, such as charts and graphs, etc. (3) The reviewers' questions and suggestion were answered point by point and supporting evidence was provided.

Our responses to reviewer’s comments are in the section below. The original reviewer comments are in black font and our responses are in red font.

Thanks for your time to handle our manuscript and we look forward to hearing from you for any further comments!

Thank you!

Sincerely yours,

Qianjing Li

Response to Reviewer Comments:

Overall comments:

This article examines the effectiveness of deep learning models in crop classification using multi-temporal remote sensing images. The authors compared and evaluated the performance of five deep learning architectures: 1D-CNNs, LSTM, 2D-CNNs, 3D-CNNs, and ConvLSTM2D, with different time-series data. The experiments analyzed the effects of temporal, spectral, and spatial information on classification accuracy and explored the most suitable multi-temporal deep learning crop classification strategies. The results indicate that combining temporal features with multi-spectral or spatial information significantly improves classification accuracy. The 3D-CNN and ConvLSTM2D models are the best deep learning architectures for multi-temporal crop classification. However, the authors suggest that the ConvLSTM architecture should be further developed for multi-temporal image crop classification.

Response: Thank you so much for your positive comments. Your suggestions and comments are very helpful to improve our manuscript. We have revised the manuscript accordingly. Please find our replies below.

Point 1: While quoting tables and figures - i preffer making them in format as Fig. and Tab. - even in brackets.

Response 1: Thank you so much for your suggestion. We are also willing to make changes. However, the citation of figures and tables used in this manuscript is strictly in accordance with the template provided in the submission system and published articles. Therefore, we did not revised in the manuscript.

Point 2: In table 2 - Please expand the descripcion.

Response 2: Thank you so much for your suggestion. We provide more description of Table 2. As follows:

(Line 157): Table 2 details the samples used for training the classification model and evaluating the accuracy. To train the model, the training and validation samples in Table 2 are randomly divided into training sample and validation sample in a ratio of 7:3.

Table 2. The five categories used in the present study for classification and the number of samples.

Sample type

Training and Validation Samples

Testing Samples

Corn

1481

4096

Soybeans

1487

4738

Spring Wheat

1445

4674

Sugarbeets

1471

4167

Others

1546

5210

Point 3: In figure 2 - i suggest adding description of a-d. even we may found it in text below.

Response 3: Thank you so much for your suggestion. We provide more description in figure 2 and the text. We explicitly list the dimensions of the vectors or matrices of the different samples. As follows:

(Line 157): The time-series classification data constructed from VI has only temporal characteristic [9], and its samples are one-dimensional vectors (Figure 2a). The time-series data con-structed directly using multi-spectral, multi-temporal images are two-dimensional ma-trices with the shape of (band, time) (Figure 2b). The time-series data constructed from VIs including the spatial neighborhood are three-dimensional matrices (Figure 2c) with the shape of (height, width, time). The multi-spectral features combined with the spatial neighborhood in multi-temporal images produce four-dimensional matrices with the shape of (time, height, width, band) (Figure 2d). The “time” in three- or four-dimensional matrices means the number of temporals in time-series.

Figure 2. Time-series samples with different dimensions. (a) 1-D time-series, (b) 2-D time-series, (c) 3-D time-series, (d) 4-D time-series.

Point 4: Figure 4 - the legend is truncated - same in fig. 5 and 6.

Response 4: Thank you so much for your suggestion. We have revised it. Figure 4 as an example.

Figure 4. Crop classification results based on VI time-series (see red boxes for more detail).

Point 5: Please provide description in methodology - what software and libraries were used for all calculations.

Response 5: Thank you so much for your suggestion. We descripted the software and libraries for data preprocessing and classification training. As follows:

(Line 137): Data preparation involved stacking and resampling the 20 m spectral bands to 10 m and removal of the coastal band, water vapor, and the cirrus band done through Sentinel Application Platform (SNAP).

(Line 258): Deep learning models were built and evaluated using the Keras library and TensorFlow. Finally, the confusion matrix and kappa coefficient from Scikit-learn are metrics for evaluating the accuracy of crop classification. The calculation of VIs and the construction of time-series data are implemented in Python.

Point 6: Fig 8 - mayby better quality?

Response 6: Thank you so much for your suggestion. The result of the revised is as follows.

Figure 8. The OA of different deep learning models.

Point 7: line 481 - after bracket - unnecessary dot.

Response 7: Thank you so much for your suggestion. We revised the sentence to make it concise.

(Line 487): Although ConvLSTM2D has fewer application in multi-temporal image crop classifica-tion than 3D-CNN [14], the results of this study show that this model approaches the classification capability of 3D-CNN.

Finally, we would like to thank the reviewer again for your comments on the manuscript!

Reviewer 2 Report

The enclosed article presented the application of different deep learning architectures for crop type classification. The purpose is to identify the better architecture with the properly selected combination of remote sensing indices. The topic can potentially attract readers in this research domain since deep learning has been intensively discussed for its capability in classification tasks. However, to me, there are critical issues that need to be addressed before the article can be considered for publication in a science journal. My major concerns are listed below for the authors' consideration.

First, the data used in the article to train the different DL models were derived from a published product developed using maximum likelihood classification. While the stated accuracy for some of the types is higher than 90% according to the product’s metadata, there are classes with accuracy lower than 50%. Further justification is needed that using such data as ground truth in the selected study area is appropriate.

Second, in addition to the accuracy of the “ground truth”, further details regarding data independence among training, evaluation and testing sets are needed to justify the performance of the models.

Third, benchmark accuracy from a traditional approach should be included, especially when the author is promoting DL method and describing them as “superior to traditional algorithms”.

Four, although I don’t feel qualified to judge the language, I see intensive editing is required to improve the result presentation and discussions.

 Below is a list of my further notes:

 L75: “superior to traditional algorithms”, please provide more references and evidence to support such a statement.

L89-91: please backup this statement with sufficient evidence in previous studies using “machine learning and traditional classification”

L97: It would help to understand why ConvLSTM is rarely used for classification with specific reasons.

L102: the “advantages of using such data” and “advantages of various deep-learning models”, according to this statement, there should be two overall objectives, one is the advantage of using S2 data and two is the advantage of using different DL models. both require an object to compare with but are not mentioned in the article.

Table 1: the impact of temporal availability on classification accuracy. The intervals between available S2 data dates varied from 5 days to ~ 1 month, will this create any artificial features learnt by the deep learning models?

L143: How is the accuracy of other crop types? When checking the Cropscape documents the accuracy for other crops/vegetables in MN 2021 is around 50-60%. Besides, the CDL maps were created with a maximum likelihood classifier, which seems less “superior” to the DL models. Please provide further justifications.

L155: further details about the distribution of the training/validation/testing pixels? What is the minimum distance between the pixels? How the pixels for each crop type are selected? Are the “validation” and “testing” pixels independent from the “training” pixels (for example, not from the same fields)?

Figure 2: Please specify what is the 4th dimension of the 4-D time series. Besides, please provide detailed figure/table captions to assist in interpreting the figures and tables.

L196: It is not clear what is the 2nd dimension of “multi-spectral”? According to L267 it seems the 2nd dimension should be “spatial information”. Also, it would be better to move Figure 2 downwards after the 1-D to 4-D has been introduced in the text and cite the figure.

L262: E4 and E5 are not “the same” as E3. Please consider revising the sentence.

L272: according to the listed experiments in Table 3 and the descriptions in the text, no tests were done with a 4-D model? If yes, I would suggest deleting the relevant descriptions about 4-D models/samples in the previous sections.

 L302 – 310: It seems the 2d-CNN model slightly improved OA by ~2%, which seems to me the additional information from the spatial aspect didn’t contribute much to the classification unless the authors can tell 2D CNN is taking more weights on spatial features? Besides, in L303, “the classification results also reveal no clear improvement in accuracy”, it is not clear what “improvement” compared to what? A benchmark model/result derived from a so-called “traditional” method might be needed here.

 L320-330: It is difficult to justify the performance of the different models presenting such small accuracy differences, even though the models are taking in different variable combinations and using different model structures. The very similar accuracies listed here might highlight the potential impacts of data independence on the experiments. The experiments were carried out in a relatively small area (approx.. 20 km x 40 km?) and the growth profiles for the same crops might be very similar across the region. this might be causing the high accuracy, and therefore the proposed experiment might not be enough to compare the real performance of the models. to compare with a benchmark model might help further clarify the improvement of DL method, however, adding diversity to the inputs (training/validation/testing) might be more important to achieve the proposed objectives.

L397-409: the average profiles are presenting the growth differences of different crops, however, this does mean that the models are learning this for classification purposes unless the authors provide the evidence (e.g. weights of trained features etc.)

L445: it is not clear to me at all what does this “indicates” mean? Does this mean the method is case specific and cannot be applied to regions with different growing situations?

Author Response

Dear Editors and Reviewers,

We would like to thank you and the reviewers for the insightful comments and suggestions to improve this manuscript. We have thoroughly revised the manuscript to address all reviewers’ comments individually. Specifically, we have revised this manuscript on the following major points. (1) We have edited and revised the language of the manuscript. (2) We checked and revised the formatting of the manuscript, such as charts and graphs, etc. (3) The reviewers' questions and suggestion were answered point by point and supporting evidence was provided.

Our responses to reviewer’s comments are in the section below. The original reviewer comments are in black font and our responses are in red font.

Thanks for your time to handle our manuscript and we look forward to hearing from you for any further comments!

Thank you!

Sincerely yours,

Qianjing Li

 Response to Reviewer Comments

Overall comments:

The enclosed article presented the application of different deep learning architectures for crop type classification. The purpose is to identify the better architecture with the properly selected combination of remote sensing indices. The topic can potentially attract readers in this research domain since deep learning has been intensively discussed for its capability in classification tasks. However, to me, there are critical issues that need to be addressed before the article can be considered for publication in a science journal. My major concerns are listed below for the authors' consideration.

First, the data used in the article to train the different DL models were derived from a published product developed using maximum likelihood classification. While the stated accuracy for some of the types is higher than 90% according to the product’s metadata, there are classes with accuracy lower than 50%. Further justification is needed that using such data as ground truth in the selected study area is appropriate.

Second, in addition to the accuracy of the “ground truth”, further details regarding data independence among training, evaluation and testing sets are needed to justify the performance of the models.

Third, benchmark accuracy from a traditional approach should be included, especially when the author is promoting DL method and describing them as “superior to traditional algorithms”.

Four, although I don’t feel qualified to judge the language, I see intensive editing is required to improve the result presentation and discussions.

Response: Thank you so much for your positive comments. Your suggestions and comments are very helpful to improve our manuscript. We have revised the manuscript accordingly. Please find our response below.

We regret there were problems with the language. The paper has been carefully revised by a professional language editing service to improve the grammar and readability.

Point 1: L75: “superior to traditional algorithms”, please provide more references and evidence to support such a statement.

Response 1: Thank you so much for your suggestion. We revised the sentence to make it concise.

(Line 76): Deep learning is a breakthrough technique in machine learning that outperforms traditional algorithms in terms of feature extraction and representation [5–7], which has led to its application in numerous RS classification tasks [8–10].

Point 2: L89-91: please backup this statement with sufficient evidence in previous studies using “machine learning and traditional classification”

Response 2: Thank you so much for your suggestion. We have revised the sentence and cited additional references.

(Line 91): For multi-temporal crop classification, both CNN and RNN provide more accurate re-sults than machine learning and traditional classification [5,9,11].

References:

  1. Cai, Y.; Guan, K.; Peng, J.; Wang, S.; Seifert C.; Wardlow, B.; Li, Z. A high-performance and in-season classification system of field-level crop types using time-series Landsat data and a machine learning approach. Remote Sens. Environ. 2018, 210, 35-47. https://doi.org/10.1016/j.rse.2018.02.045.
  2. Zhong, L.; Hu, L.; Zhou, H. Deep learning based multi-temporal crop classification. Remote Sens. Environ. 2018, 221, 430-443. https://doi.org/10.1016/j.rse.2018.11.032.
  3. Ji, S.; Zhang, C.; Xu, A.; Shi, Y., Duan, Y. 3D convolutional neural networks for crop classification with multi-temporal remote sensing images. Remote Sens. 2018, 10(1), 75. https://doi.org/10.3390/rs10010075.

Point 3:L97: It would help to understand why ConvLSTM is rarely used for classification with specific reasons.

Response 3: Thank you so much for your suggestion. We revised the sentence to make it concise.

(Line 99): However, due to the prevalence of CNNs and RNNs and the requirement for higher da-ta dimensions, the ConvLSTM model is less commonly used in multi-temporal crop classification. Nevertheless, the potential of the ConvLSTM model deserves further ex-ploration.

The reasons mentioned in the comment are as follows: (1), There are so many deep learning models are proposed in other research fields. Individual models are less likely to be applied to a specific application domains, especially some variants of deep learning model. (2), It is clear that CNN and RNN models achieve satisfactory classification results in multi-temporal crop classification. And this also reduces the need for other deep learning models in research and ignores the possibility of other model applications. (3), Higher requirements for data dimensionality (4D or higher). The model's internal computational are not met by the lower data dimensions.

4D means the data matrix with shape of (time, height, width, band).

Point 4: L102: the “advantages of using such data” and “advantages of various deep-learning models”, according to this statement, there should be two overall objectives, one is the advantage of using S2 data and two is the advantage of using different DL models. both require an object to compare with but are not mentioned in the article.

Response 4: Thank you so much for your comment. We revised the manuscript to make it concise.The “advantages of using such data” is described in lines 68-74. This part shows the advantages of multi-temporal images with multi-spectral, multi-temporal, and multi-spatial. And Sentinel-2 images also have these advantages and were selected as research data for this work.

The “advantages of various deep-learning models” is described in lines 76-100. This part presents the structural advantages of deep learning models to meet the applications of different classification data.

The description of line 102 is derived from the summary of the previous content.

Point 5: Table 1: the impact of temporal availability on classification accuracy. The intervals between available S2 data dates varied from 5 days to ~ 1 month, will this create any artificial features learnt by the deep learning models?

Response 5: Thank you so much for your question.

We all know that weather and external conditions such as clouds have a great influence on the acquisition of remote sensing images. so did Table 1.

[8] pointed out that when the time series is short or the temporal density is sparse, deep architecture exerts little effect on classification improvements. It is not clear to what extent the different time intervals affect the classification accuracy. And it will serve as a research point for our next article.

Interpretability for deep learning models internally is always a challenge. It is still worth exploring how to parse out some artificial features from the deep learning models. And we will also conduct research on this question.

References:

  1. Dou, P.; Shen, H.; Li, Z.; Guan, X. Time series remote sensing image classification framework using combination of deep learning and multiple classifiers system. International Journal of Applied Earth Observation and Geoinformation. 2021, 103, 102477. https://doi.org/10.1016/j.jag.2021.102477.

Point 6: L143: How is the accuracy of other crop types? When checking the Cropscape documents the accuracy for other crops/vegetables in MN 2021 is around 50-60%. Besides, the CDL maps were created with a maximum likelihood classifier, which seems less “superior” to the DL models. Please provide further justifications.

Response 6: Thank you so much for your comment. We revised the text to make it precise.

(Line 147): Therefore, a result of visual interpretation of multi-temporal Sentinel-2 images based on CDL data was used to select the primary crop samples for training and testing our crop classification model.

Firstly, we find the classification accuracy of several crops in Minnesota (Table 1). The accuracy in Table 1 is significantly higher than 60%.

Table 1 Accuracy of 2021 Minnesota Cropland Data Layer

Type

Producer's Accuracy

User's Accuracy

Kappa
Corn 94.7% 96.9% 0.929
Soybeans

95.0%

94.5%

0.934

Spring Wheat

94.3%

93.5%

0.941

Sugarbeets

93.1%

98.3%

0.930

In addition, the CDL data in the manuscript were only used as reference data to assist us in selecting the sample. The samples in the study area were generated by visual interpretation of multi-temporal sentinel image based on CDL to ensure as much correctness and accuracy as possible.

Point 7: L155: further details about the distribution of the training/validation/testing pixels? What is the minimum distance between the pixels? How the pixels for each crop type are selected? Are the “validation” and “testing” pixels independent from the “training” pixels (for example, not from the same fields)?

Response 7: Thank you so much for your comment. We revised the manuscript to make it concise after line 152.

The samples are generated for model training and accuracy evaluation, respectively. The samples for model training were created from Creat Random Points in Arcgis to ensure distributed throughout the study area. Minimum allowed distance of 100 m is set to ensure the independence between different sample points. The samples for accuracy evaluation were also generated based on the above description.

The training samples and validation samples are independent of each other. They are randomly divided in the ratio of 7:3 from the samples used for model training.

The training samples and testing samples come from two different sample processes, and we ensure that there are no human subjective factors interfering in the sample generation process, and they can be assumed to be independent of each other.

Point 8: Figure 2: Please specify what is the 4th dimension of the 4-D time series. Besides, please provide detailed figure/table captions to assist in interpreting the figures and tables.

Response 8: Thank you so much for your suggestion. We revised the text to better explain the dimensionality of the time-series data in Figure 2. As follows:

(Line 200): The time-series classification data constructed from VI has only temporal characteristic [9], and its samples are one-dimensional vectors (Figure 2a). The time-series data con-structed directly using multi-spectral, multi-temporal images are two-dimensional ma-trices with the shape of (band, time) (Figure 2b). The time-series data constructed from VIs including the spatial neighborhood are three-dimensional matrices (Figure 2c) with the shape of (height, width, time). The multi-spectral features combined with the spatial neighborhood in multi-temporal images produce four-dimensional matrices with the shape of (time, height, width, band) (Figure 2d). The “time” in three- or four-dimensional matrices means the number of temporals in time-series.

Figure 2. Time-series samples with different dimensions. (a) 1-D time-series, (b) 2-D time-series, (c) 3-D time-series, (d) 4-D time-series.

Point 9: L196: It is not clear what is the 2nd dimension of “multi-spectral”? According to L267 it seems the 2nd dimension should be “spatial information”. Also, it would be better to move Figure 2 downwards after the 1-D to 4-D has been introduced in the text and cite the figure.

Response 9: Thank you so much for your comment. We revised the manuscript to make it concise after line 199.

The dimension has no order in this manuscript, it only represents different shapes of the samples data with one-dimensional vector (1D), two-dimensional matrices (2D), three-dimensional matrices (3D) and four-dimensional matrices (4D). The description of the samples with different dimensions corresponding to different deep learning models can be found after line 213.

Point 10: L262: E4 and E5 are not “the same” as E3. Please consider revising the sentence.

Response 10: Thank you so much for your suggestion. We revised the sentence to make it concise.

(Line 269): E4 and E5 add shortwave infrared (SWIR) and red-edge spectral bands to E3, respectively.

Point 11: L272: according to the listed experiments in Table 3 and the descriptions in the text, no tests were done with a 4-D model? If yes, I would suggest deleting the relevant descriptions about 4-D models/samples in the previous sections.

Response 11: Thank you so much for your suggestion. We expanded the contents of Table 3 to better explain the correspondence between the model and the data.

Table 3. Experiment groups.

Number

Features

Samples Dimensions

Model

E1

NDVI

1-D time-series

1D-CNN

LSTM

E2

EVI

E3

B2348

2-D time-series

1D-CNN

LSTM

E4

B2348+B11+B12

E5

B2345678

E6

All Bands

E7

NDVI

3-D time-series

2D-CNN

E8

EVI

E9

B2348

4-D time-series

3D-CNN

ConvLSTM-2D

E10

All Bands

Point 12: L302 – 310: It seems the 2d-CNN model slightly improved OA by ~2%, which seems to me the additional information from the spatial aspect didn’t contribute much to the classification unless the authors can tell 2D CNN is taking more weights on spatial features? Besides, in L303, “the classification results also reveal no clear improvement in accuracy”, it is not clear what “improvement” compared to what? A benchmark model/result derived from a so-called “traditional” method might be needed here.

Response 12: Thank you so much for your comment.

As we can see that 2D-CNN model slightly improved OA by ~2% in this manuscript. However, it is undeniable that 2D-CNN improves the classification accuracy. In addition, the contribution of spatial information to eliminate salt-and-pepper noise cannot be ignored (Figure 4).

We revised sentence to make it precise.

(Line 310): There is no significant difference between the classification accuracy of LSTM and 1D-CNN models in E1, and the same is true in E2. This explains the difficulty of improving accuracy using only time-series data constructed from a single VI.

This manuscript focuses on the issue of applying multiple deep learning models with multi-temporal images. Deep learning methods outperform machine learning and traditional methods in numerous RS classification tasks. This part compares the effects of temporal and spatial information on the classification results. This work sets up multiple sets of comparison experiments (Table 4) in which the classification results of different time-series data (NDVI or EVI) and models (LSTM, 1D-CNN or 2D-CNN) combined can be used as a benchmark for each other. Therefore, a benchmark model/result derived from traditional method was not selected in this manuscript to draw the same conclusions as the above statement again.

Table 4. Classification accuracy produced by various models with VI time series.

Number

Model

Accuracy

OA

Kappa

E1

1D-CNN

92.50

0.906

LSTM

93.25

0.915

E2

1D-CNN

92.76

0.909

LSTM

93.94

0.924

E7

2D-CNN

94.74

0.934

E8

2D-CNN

94.76

0.934

Point 13: L320-330: It is difficult to justify the performance of the different models presenting such small accuracy differences, even though the models are taking in different variable combinations and using different model structures. The very similar accuracies listed here might highlight the potential impacts of data independence on the experiments. The experiments were carried out in a relatively small area (approx.. 20 km x 40 km?) and the growth profiles for the same crops might be very similar across the region. this might be causing the high accuracy, and therefore the proposed experiment might not be enough to compare the real performance of the models. to compare with a benchmark model might help further clarify the improvement of DL method, however, adding diversity to the inputs (training/validation/testing) might be more important to achieve the proposed objectives.

Response 13: Thank you so much for your comment.

Firstly, we emphasize that the difference for classification accuracy does not only depend on the performance of different models. The difference of classification input data in experiments of different models cannot be ignored. As described in 3.2 of the manuscript for the sample dimensions, the different input data present the degree of utilization of multi-spectral, multi-spatial and multi-temporal information in time-series remote sensing images. And the degree of using these information is the main reason to improve the classification accuracy. Of course, we firstly assume that the deep learning model can fully exploit the effective information in the classification input data.

Small fluctuations in accuracy are inevitable in the results of repeated experiments (Figure 8) and different models (Table 4), which may be due to various factors, such as the degree of training of the model and model structure, etc. As mentioned above, the comparison of results shows the contribution of different input data to the accuracy. Smaller areas and similar growth conditions for the same crops can avoid interference, such as misalignment of planting time, are more favorable for work to study the effects of multi-spectral, multi-temporal and multi-spatial information on classification accuracy.

Many studies have shown the fact that deep learning methods outperform machine learning and traditional methods in remote sensing image classification tasks [5, 6, 9,12]. We agree with this statement and research on this basis. Therefore, traditional or machine learning methods were not selected as benchmark to draw the same conclusions as the above statements again in this manuscript. In addition, we focus on the problem of application between deep learning models and time-series data and the impact of deep learning models and the degree of use of time-series images on classification accuracy.

The idea that adding diversity to the inputs (training/validation/testing) might be more important is something we can all agree on. We will pay more attention to the deficiency in this point in the subsequent study.

References:

  1. Cai, Y.; Guan, K.; Peng, J.; Wang, S.; Seifert C.; Wardlow, B.; Li, Z. A high-performance and in-season classification system of field-level crop types using time-series Landsat data and a machine learning approach. Remote Sens. Environ. 2018, 210, 35-47. https://doi.org/10.1016/j.rse.2018.02.045.
  2. Belgiu, M.; Csillik, O. Sentinel-2 cropland mapping using pixel-based and object- based time-weighted dynamic time warping analysis. Remote Sens. Environ. 2018, 204, 509-523. https://doi.org/10.1016/j.rse.2017.10.005.
  3. Zhong, L.; Hu, L.; Zhou, H. Deep learning based multi-temporal crop classification. Remote Sens. Environ. 2018, 221, 430-443. https://doi.org/10.1016/j.rse.2018.11.032.
  4. Qu, Y.; Yuan, Z.; Zhao, W.; Chen X.; Chen, J. Crop classification based on multi-temporal features and convolutional neural network. Remote Sensing Technology and Application. 2021, 36(2), 304-313. Doi: 10.11873/j.issn.1004‐0323.2021.2.0304.

Point 14: L397-409: the average profiles are presenting the growth differences of different crops, however, this does mean that the models are learning this for classification purposes unless the authors provide the evidence (e.g. weights of trained features etc.)

Response 14: Thank you so much for your comment. This is closely related to the next research work.

The experimental results are obtained by training the models several times. And the classification task of deep learning is done in this way in most studies. However, the lack of interpretation is regarded as a major drawback of deep learning models. This situation also appears in this manuscript. The differences in the average profiles indicate the potential importance of these input characteristics in differentiating crops. How to interpret the learning and representation of the spectral differences of different crops in the deep learning models is a difficult problem, which is the focus of our subsequent research.

Point 15: L445: it is not clear to me at all what does this “indicates” mean? Does this mean the method is case specific and cannot be applied to regions with different growing situations?

Response 15: Thank you so much for your comment. We revised the manuscript to make it concise.

(Line 452): Unlike the result that 1D-CNN accuracy is higher than that of LSTM [9], increasing the number of spectral bands in this work causes the crop classification accuracy of 1D-CNN close to that of LSTM. This indicates that classification data and application scenarios may also affect the accuracy of the classification.

The deep learning models in the manuscript are all available for multi-temporal crop classification. And these models can also be applied to regions with different growing situations.

Finally, we would like to thank the reviewer again for your comments on the manuscript!

Round 2

Reviewer 2 Report

I quickly checked the authors' response. They had revised the texts to some of my specific comments, however not willing to address the main issues of my concern. They said in the response "for next article" or "for future research", to me this is not the way for a scientific article to be published. I appreciate the authors' great efforts explaining their ideas in the response letter, however this is what needed inside the article rather.